# Health Psychology Services for People in Disadvantaged Regions of Hungary: Experiences from the Primary Health Care Development Model Program

**DOI:** 10.3390/ijerph20053900

**Published:** 2023-02-22

**Authors:** Viola Sallay, Tamás Martos, Lilla Lucza, Orsolya Papp-Zipernovszky, Márta Csabai

**Affiliations:** 1Institute of Psychology, University of Szeged, 6720 Szeged, Hungary; 2Doctoral School of Education, University of Szeged, 6720 Szeged, Hungary; 3Institute of Psychology, University of the Reformed Church, 1091 Budapest, Hungary

**Keywords:** community health psychology, primary health care, disadvantaged populations, well-being, integrated care

## Abstract

Background: The importance of community health psychology in providing complex bio-psycho-social care is well documented. We present a mixed-method outcome-monitoring study of health psychology services in the public-health-focused Primary Health Care Development Model Program (2012–2017) in four disadvantaged micro-regions in northeast Hungary. Methods: Study 1 assessed the availability of the services using a sample of 17,003 respondents. Study 2 applied a follow-up design to measure the mental health outcomes of the health psychology services on a sample of 132 clients. In Study 3, we conducted focus-group interviews to assess clients’ lived experiences. Results: More mental health issues and higher education predicted a higher probability of service use. Follow-up showed that individual and group-based psychological interventions resulted in less depression and (marginally) higher well-being. Thematic analysis of the focus-group interviews indicated that participants deemed topics such as psychoeducation, greater acceptance of psychological support, and heightened awareness of individual and community support important. Conclusions: The results of the monitoring study demonstrate the important role health psychology services can play in primary healthcare in disadvantaged regions in Hungary. Community health psychology can improve well-being, reduce inequality, raise the population’s health awareness, and address unmet social needs in disadvantaged regions.

## 1. Introduction

The populations of countries with a strong primary health care system tend to have better health status, fewer hospital admissions, and less socioeconomic health inequality [1,2,3,4]. This worldwide trend parallels the development and expansion of the scope of primary health care activities oriented towards prevention. Prevention can be achieved by transposing specialist medical services into primary care and connecting primary care with community services. This shift may involve patients in treatment and health management [5], decreasing health inequality.

The paper presents the results of a monitoring program of the health psychology services in the public-health-focused Primary Health Care Development Model Program, introduced to the most disadvantaged regions of Hungary. The community-health-focused program aimed to improve the health of the disadvantaged population by providing new preventive services within the primary care setting. Beyond the involvement of dietitians, physiotherapists, and public health professionals, the program employed community health psychologists. The present report focuses on experiences with the community health psychology services.

### 1.1. The Primary Care Development Model in Hungary

Health systems worldwide are facing numerous challenges due to the changing demographic, socio-cultural, and natural/technological environment, and Hungary is no exception. In a European comparison, Hungary is amongst those countries with less favorable indicators for lifestyle-related risk factors, chronic non-communicable diseases, and deaths. Moreover, the emigration of health professionals and the aging of medical society threaten the health system’s sustainability [6,7,8]. In addition, the perception of psychological problems that is prevalent in Hungarian society is deficient, and awareness of the importance of prevention and the danger of unhealthy behaviors is low [9]. It is therefore important that health professionals become more knowledgeable about the psychosocial aspects of illness and how they can help people reduce these risks. Among many healthcare workers, there is a lack of understanding of the factors that influence mental and physical health and the necessary organizational background for the provision of effective and equitable health services.

With all these challenges in mind, an innovative pilot program, the Primary Care Development Model Program, was implemented in four of the most disadvantaged regions of Hungary (in settlements with populations partly living in poverty) within the framework of the “Swiss-Hungarian Cooperation Programme” between 2013 and 2017 [10]. The Model Program was a community-based health-focused program designed to improve the population’s health by providing primary patient care and emphasizing the use of disease prevention programs, screenings, counseling, and health-promotion services [11,12]. In addition, the Model Program was designed with particular attention to the Roma minority, in line with the agendas of European health policies [13].

Four functionally enhanced and reorganized primary care centers—so-called general practitioner clusters (GP clusters)—were set up. The four GP clusters offered new services to the local population, including health-status surveys, lifestyle counseling, physiotherapy, preventive services, dietary and health-psychology counseling, and community health-promotion programs. The services involved dieticians, physiotherapists, psychologists, public health professionals, and a new type of helping professional called the Roma “health mediator” [14]. Health mediators were recruited from the Roma population and trained for their job within the frame of the program. Their role was especially important for the population-wide promotion of the services. In addition, one of the essential service areas of the program was the employment of community health psychologists [12,15].

### 1.2. Community Health Psychologists’ Contribution to Primary Care

New models of primary care aim at strengthening its community function, in line with the claim that supporting patient integration into society increases the effectiveness of care. At the same time, treating mental health problems has become a significant part of the workload of primary care professionals. These shifts in approach and challenges facilitate the integration of mental health services and home and social care. Moreover, they create an opportunity to involve professions that are traditionally not part of primary care, such as psychologists, physiotherapists, and dietitians. Evidence demonstrates that patients supported by mental health professionals recover sooner, and their conditions are more sustainable than if they are solely treated by general practitioners (GP). Additionally, the number of GP consultations significantly decreases if mental health practitioners work in practices [16].

Gunn and Blount proposed a model whereby psychologists collaborate with primary care medical teams [17]. This integrated, coordinated model allows psychologists to treat patients, including the elderly, those with somatic complaints, and chronic patients who would otherwise have difficulty accessing specialists. The essence of the integrated model is the community health psychologists’ ongoing consultation and communication with doctors within interprofessional teams [17]. Haas and Degruy argued similarly for the involvement of psychologists in primary care. They found that primary care patients have various psychological needs that can be classified into three general categories: psychopathology and mental disorders; stress-related symptoms and problems with chronic illnesses or behavioral health problems; and vulnerable groups (e.g., victims of abuse and socially or economically isolated persons). In addition, patients in primary care settings (especially those with mental health problems) are likely to be older, less well educated, poorer, and members of a minority group [18]. However, Thielke and his colleagues also identified challenges that need addressing in this field, including defining health psychologists’ competences in primary care and clarifying their role in integrated care [19].

Beyond offering primary health benefits, community health psychology services can help reduce harmful social inequalities by working with a broad definition of health, namely a state of mental, physical, and social well-being [20,21]. Community health psychology has been described as “a body of theory and practice that focuses on the processes of collective action through which communities collectively identify the impacts of oppressive social relations on their wellbeing and engage in social struggles to create more health-enabling social environments” [22]. This definition implies the “coming of age” of community health psychology and demonstrates its potential to promote health through collective action in local settings.

### 1.3. The Present Research

The main aim of the present research is to monitor the characteristics and outcomes of the health psychology services of the Primary Health Care Development Model Program in Hungary. The Model Program introduced a new primary care model to four disadvantaged regions of Hungary, involving integrating community health psychologists into GP clusters. According to the Operations Manual of the Practice Teams [23], the roles of a health psychologist are (1) to provide individual and group therapy sessions; (2) to provide tailor-made lifestyle counseling in areas such as stress management, weight-loss programs, and quitting smoking; (3) to participate in the rehabilitation of patients with somatic diseases; and (4) to participate in prevention and screening programs. Therefore, we explored the following research questions concerning the health psychology services in the GP clusters.

First, what are the characteristics of the clients who had access to the health psychology services compared to those of the general population in the regions affected by the model program (Study 1)? The answer to this question can shed light on the contribution of health psychology services to increasing equitable healthcare access in disadvantaged subpopulations. Specifically, we tested whether service use was more likely among subpopulations in need.

Second, what service outcomes can be measured in terms of mental health indicators (Study 2)? Understanding this is important because one of the main avenues for empowering people in need is improving their mental health [22]. We assumed that participants’ mental health and well-being would significantly improve during the time of use of health psychology services.

Third (a qualitative form of inquiry), we wanted to gather information about the participants’ subjective experiences and opinions about the health psychology services (Study 3). Personal experiences can provide valuable feedback that helps programs function more sustainably and gives voice to a vulnerable and otherwise unheard population. Moreover, these experiences may contribute to understanding the program’s strengths and areas for improvement.

We present these surveys as three interconnected studies, each containing a detailed overview, a description of the methods and results, and a brief discussion. Finally, we summarize and interpret the findings in a general discussion.

## 2. Study 1

### 2.1. Overview

The aim of Study 1 was to explore the characteristics of those participants who had access to health psychology services in the aforementioned disadvantaged regions of Hungary. The Primary Health Care Development Model Program introduced a new community-based screening procedure, the Health Status Assessment (HSA) [24], which aimed to facilitate the in-time recognition of exposure to avoidable (e.g., lifestyle-related) risk factors and early-stage disorders through screening. The HSA database also made it possible for us to identify which people were more likely to receive health psychology services by comparing the characteristics of the service users (see Study 2 for a detailed description) and the general population, as represented by the HSA data. We assumed that participants’ service use would be affected by their gender, ethnicity, age, level of education, and general and mental health status [25,26,27]. Accordingly, we tested whether these variables could predict psychological service use.

### 2.2. Methods

#### 2.2.1. Participants and Procedure

The HSA was coordinated by the screening teams of the GP clusters (including the community nurses, the public-health specialists, and the GPs). The screening procedure was carried out by health mediators whose task was to connect healthcare providers with the local community, focusing on adults with a disadvantageous socioeconomic status [28]. During the monitoring period (from November 2013 to March 2016), of the total number of adults registered in the GP clusters, 22,652 adults took part in HSA, which translates into a 70% participation rate. From the whole HSA sample, 17,003 adults (7.4% Roma minority) had complete response sets, 107 of whom (0.6%) also participated in the intervention outcome-monitoring study and received health psychology services (2.8% Roma minority) (see Study 2). By connecting data from the HSA assessment (demographic and mental health data) and the outcome-monitoring study (Study 2: participant or not), we could estimate the predictors of access to health psychology services. Therefore, we used the characteristics available in the HSA and considered them relevant to treatment entry.

#### 2.2.2. Materials

We used the following variables from the HSA measurement scheme as predictors in their original form or as recoded variables. Gender (male/female); age (18–24/25–44/45–64/65+); education (categories: maximum eight years of primary school/secondary without graduation/secondary with graduation/higher education); subjective financial situation (very bad, bad, optimal, good, excellent); Roma identity (no/yes); and BMI (not overweight, overweight, or obese). The HSA measurement included systolic blood pressure (normal/high), diastolic blood pressure (normal/high), and information on smoking status (non-smoker, smoker).

Mental health status was measured using the Beck Depression Inventory—Short Version (BDI-S) [29], a commonly used nine-item questionnaire for assessing depressive symptomatology in community-based surveys, and the General Health Questionnaire 12 (GHQ-12) [30], a self-administered questionnaire for screening psychiatric symptoms in the general population. BDI-S items present the physical, cognitive, and emotional symptoms of depression with response options ranging from 0 = “not at all” to 3 = “most of the time” (summarized scores 0–9: normal/10–18: mild depression, 19–24: moderate depression/25+: severe depression). Response options for GHQ-12 range from 0 = “not at all” to 3 = “most of the time” (0–4: normal/5+: high).

Subjective health status was measured with the one-item self-rated health (SRH) estimate (very bad, bad, optimal, good, excellent). Health locus of control was measured with the following question: How much can you do for your health? (very much/much/little/nothing).

#### 2.2.3. Statistical Procedures

We ran a multivariate binary logistic regression analysis to assess the predictive power of various characteristics in relation to the receipt of health psychology services. Multivariate binary logistic regression analysis is appropriate to detect the independent predictive power of several particular factors in the outcomes studied. In our case, the outcome measure was inclusion (vs. non-inclusion) in the outcome-monitoring study, and the predictors were the characteristics assessed in the HSA. The alpha level for significance testing was set to *p* = 0.05 for the subsequent analyses.

### 2.3. Results

The model explained 12.0% of the variance (Nagelkerke’s R^2^; chi-square (df = 37) = 157.99, *p* < 0.001). Among the sociodemographic variables, education was the most important predictor of participation in the intervention outcome-monitoring study. Compared to those with a maximum of eight years of primary school education, those who had completed secondary school but without graduating were more than four times, those who had graduated more than six times, and those with a higher-level education more than eight times more likely to have participated in the outcome-monitoring study. The other significant predictive sociodemographic variable was gender: women were three times more likely to have participated in the outcome-monitoring study. Finally, among the mental health characteristics, the BDI-S score and the GHQ12 score were significant predictors: having a higher BDI-S score (moderate or severe depression) and a higher GHQ12 score (depression, anxiety, social dysfunction, self-esteem) predicted individuals’ involvement in the outcome-monitoring study. The results are shown in Table 1.

### 2.4. Short Discussion

Our results show that poor mental health (e.g., depression and anxiety) was associated with a higher likelihood of use of health psychology services. There was a tendency for health psychology services to be sought out primarily by those whose health conditions warranted this. Besides this, some of our findings supported the results of previous studies, namely that specific sociodemographic characteristics (such as gender and higher social status) are significant predictors of engagement with health psychological services [25,26,27]. Being female and having a higher education was linked to a higher probability of access to the service. Other characteristics such as age and ethnic background (i.e., Roma identity) were not significant predictors, indicating balanced access to the services in this regard. It is important to note, however, that supporting the access of the Roma population was an explicit aim of the model program [12,14]. Therefore, their proportional representation among service users indicates a challenge that needs consideration even if the deviation was not significant. More explicitly, the results confirmed that other social groups, such as males and participants with a lower level of education, need more support to find their way to services that are available [31,32].

## 3. Study 2

### 3.1. Overview

Integrated models of primary healthcare that include health psychology services are cost-effective and support the physical and mental health of patients of all ages [33,34,35,36,37,38,39]. During the last year of the Model Program, we implemented a follow-up questionnaire assessment to monitor the outcomes of the health psychology services provided within the frame of the Model Program. We assumed an increase in well-being and a decrease in depression among recipients of health psychology services.

### 3.2. Methods

#### 3.2.1. Participants and Procedure

With the help of the health psychologists working in the Model Program, we recruited volunteer participants from among the patients who had received health psychology services individually or in groups during the assessment period. The first assessment was recorded for each participant before starting the actual health psychology service (T1). This assessment, which constituted a separate phase from the HSA, provided the baseline values for the outcome-monitoring study. Participants were also assessed at the end of the service (T2), which provided an opportunity to estimate the extent of the mental health change during the program compared to the baseline values. The average time between the two assessments was about 120 days.

Regarding our sample, at T1, 156 participants completed the questionnaire, while at the second measurement point (T2), 137 of them did so (dropout rate: 12.2%). Of the 156 T1 participants, 107 had previously participated in the HSA study, as described in Study 1. However, HSA variables and data were not analyzed in this study. The basic characteristics of the sample are shown in Table 2.

#### 3.2.2. Materials

The outcome-monitoring study’s baseline and follow-up questionnaires were separate, abbreviated, and, in certain aspects, extended variants of the HSA assessment tool. Both the baseline (T1) and the follow-up (T2) questionnaire package included the following instruments: Beck Depression Inventory—Short Version (BDI-S) (see Study 1) [29] and the WHO Well-Being Index (WHO-WBI) [40], a five-item questionnaire for assessing the experience of positive emotional states, with response options ranging from 1 = “not at all true” to 5 = “completely true”. The questionnaire also included questions about socio-demographic characteristics (e.g., age, gender, and education).

### 3.3. Results

To test for potential improvements in mental health indicators, we compared pre- and post-intervention BDI-S and WHO-WB scores. Both indicators improved during the interventions: on average, participants had fewer depressive symptoms and experienced positive moods and feelings more frequently. Preliminary paired-sample *t*-tests showed significant differences between measurement points for both indicators. In the next step, we applied repeated measures ANOVA and controlled for gender, age, education level, and BMI to test the associations for potential confounding factors. Concerning BDI-S, ANCOVA with Greenhouse–Geisser correction was significant and showed differences between the measurement points of BDI-S (F(1, 131) = 4.520, *p* < 0.035). Regarding the WHO Well-Being Index, the ANCOVA with Greenhouse–Geisser correction showed no significant differences between the measurement points of the WHO Well-Being Index (F(1, 131) = 2.315, *p* < 0.131). Results are presented in Table 3.

### 3.4. Short Discussion

Results of the outcome-monitoring study show that health psychological interventions may have beneficial effects on participants’ mental health, which supports the results of several previous studies [33,34,35,36,37,38,39]. This potential effect was robust in the case of depressive symptoms: the improvement was verifiable even after controlling for potential confounding variables. In contrast, while improvement in well-being was significant in bivariate analysis, the control variables partly explained the change in scores. This association means that multivariate analysis did not identify specific mental health improvements for participants. However, the potential of integrating health psychology services into primary care to achieve better mental health and well-being (fewer depressive symptoms and more frequent positive emotions) is supported by our results [18,19]. Later programs should be better tailored to participants’ personal needs, including the resources and risks in their sociodemographic background. Moreover, the present results support the inclusion of routine outcome monitoring [41,42] in service protocols.

## 4. Study 3

### 4.1. Overview

In addition to the quantitative assessments (Study 2), we organized focus groups with the participants and conducted semi-structured interviews with them. Focus-group interviews are deemed appropriate when the purpose of a study is to explore people’s shared and specific experiences and knowledge about a topic [43]. We aimed to explore the participants’ shared and specific experiences and opinions about health psychology services. The inclusion of these experiences into program evaluation may provide valuable feedback on factors otherwise unaddressed, such as the strengths and underdeveloped aspects of a program. Moreover, they may help with identifying a hitherto unknown but vulnerable population—in our case, the participants of health psychology services in a disadvantaged region.

### 4.2. Methods

#### 4.2.1. Participants and Procedure

We employed purposeful sampling adapted to the research question; therefore, we recruited participants with experience with health psychology services as clients. The selection criteria were as follows: age above 18 years, former or ongoing experience with health psychology services, and a willingness to share lived experiences of the health psychological consultation process (irrespective of the outcome of the process). During the inductive analysis of the interviews, we used the method of continuous comparison and built a matrix of emergent themes. Interviews ranged from 70 to 120 min in length. All interviews were audio-recorded, anonymized, and then literally transcribed. In cooperation with health psychologists and public health coordinators, we organized one focus group for each model GP cluster with patients receiving health psychological services (2–8 patients per group). A total of 21 patients participated in four focus-group interviews. The focus groups were led by two group leaders (the first and second authors) based on the interview guide compiled for this purpose.

#### 4.2.2. Materials

A detailed interview guide was developed for the focus-group interviews. The interview guide emphasized the importance of exploring subjective experiences, thoughts, feelings, and social processes. In addition, it covered the following general topics: (1) the process of referral to and usage of the health psychology service, (2) the broader social context of the use of the health psychology service, and (3) changes experienced by participants.

#### 4.2.3. Analytical Procedure

We qualitatively analyzed the focus-group interviews to answer the following research question: What are the participants’ experiences with the health psychology interventions? The data collection and analytical procedure for the focus-group interviews was developed by combining the constant comparison method of grounded theory methodology [44] and framework analysis (FwA) [45]. FwA “is explicitly geared towards generating policy- and practice-orientated findings…” [46].

### 4.3. Results

Focus-group members experienced positive mental health developments on individual and community levels. Finally, we present the five main themes that emerged in the analysis and present interview quotes characteristic of each theme.

#### 4.3.1. Theme 1—Community Instead of Isolation

One of the focus groups’ most powerful and recurring themes was the shared experience that the health psychology services provided a pathway to new social relationships. Elderly clients and clients living with chronic diseases experienced a significant change in their relationships. Even clients living alone or with a relative suffering from chronic illness experienced a sense of community and belongingness due to the regular group activities led by the health psychologists:


*And now there is an opportunity, there is a community where they don’t just listen to you, but you even get help, helping sentences, or you see the interested look on the face of the other person, that they don’t just listen, but also care about my problem. They may give advice based on their own experience. (Client)*


The quotation indicates that peers’ shared experiences can be as empowering as professional advice. Other participants in individual settings emphasized the novelty of “speaking out” problems to another person (the health psychologist) after years of “keeping them in”.

#### 4.3.2. Theme 2—Autonomy and Activity

The experience of the health psychology consultations compensated participants for a feeling of vulnerability due to deprivation, illness, and low mood. They talked about newly acquired skills that helped them gain more control over their lives. Moreover, these experiences helped motivate the participants to address and involve others in the program:


*I think it would be important for all of us to pass it on and say: ‘yes, this is good, not just because we’re happy and having a good time, but because it takes us further in life, in our attitude towards life.’ (Client)*


Changes toward a more autonomous and active way of living encompassed new forms of relationship maintenance with relatives and close ones. As one participant put it, “She [the health psychologist] helped me to start over with walking and, later, traveling. Now I’m preparing to visit my son in the capital”. This activity was acknowledged by the focus group for its boldness (going into the metropolis from the village) and, at the same time, for the hoped-for emotional closeness.

#### 4.3.3. Theme 3—Openness to Psychological Help and Positive Thinking

An awareness and acceptance of psychological help can become the norm in small communities where this was previously rejected and considered shameful. Additionally, clients’ horizons could be broadened through psychoeducation and self-experience:


*We are living now! …a new world has opened up! And it’s such a beautiful thing when a psychologist leads you to find the beauty of your living now. (Client)*


Other participants also referred to “new ways of thinking” and the “unexpected usefulness” of psychological counseling that they could implement in their lives.

#### 4.3.4. Theme 4—Healing Grief and Negative Emotions

A recurring mental health problem in the studied groups was the devastating effect of untreated, unelaborated loss and grief on the quality of life and health. This burden was sometimes significantly reduced with the help of psychological interventions; participants reported getting over grief and loss. For example, a participant who had been mistreated during eye surgery talked about her subsequent recovery:


*…when I started coming here, there was such terrible hatred, anger, and other kinds of negative feelings in my soul towards those who messed me up in the hospital… [but] it disappeared from me, and I put things in a different frame, and that’s what the psychologist taught me. (Client)*


This quotation shows that while healthcare experiences can be a recurring source of stress and trauma, health psychologists’ consultations have mitigating and empowering potential. Self-acceptance and self-compassion were other essential skills that many participants mentioned: in disadvantaged regions, people often feel guilty and ashamed about their situation. The health psychologists’ acceptance and normalization of negative feelings contributed to a new way of thinking, as mentioned above.

#### 4.3.5. Theme 5—The Security of Mental and Somatic Rehabilitation

In addition to the above experiences, patients reported better cooperation (adherence) with their general practitioners (GPs). In the patients’ experience, health psychologists provided valuable insights into mental and somatic rehabilitation:


*I [had recently been] operated on and [experienced] all this and other bad things… And then the doctor saw this in my final report, and he told the psychologist, and here, my rehabilitation practically continued with this program.… Here, however, I found a complete cure. First, I was with the psychologist for six sessions, then we asked for six more. (Client)*


Cooperation between health professionals—here, between the GP and the health psychologist and in other cases between the psychologist and the dietician or the physiotherapist—provided the participants with a valuable experience of security. The psychological help was thus integrated into the team collaboration process, giving the participants a sense of “completeness” in their care.

### 4.4. Short Discussion

The analysis of the interviews revealed the impact of the health psychologists’ work from the patients’ perspective. The themes represent the main paths the patients travelled on toward improved physical and mental well-being. They showed that positive resource-mobilizing and mental health processes were taking place in the communities under study at the individual and community levels. An important experience was a decline in feelings of isolation and the experience of peer support among elderly clients and clients with chronic illnesses (Theme 1). Based on all previous findings, reducing isolation and increasing support is one of the most important forms of help for disadvantaged groups [47]. Changes in social experiences, the participants’ increased self-confidence and activity (Theme 2), and a broadening of horizons through psychoeducation (Theme 3) are mutually supportive phenomena. A significant result is the recognition and acceptance of psychological support as normal in communities where it was previously associated with shame and rejected. A recurring theme in the studied groups was the detrimental impact of unaddressed and unprocessed loss and bereavement processes on the quality of life and mental health (Theme 4). Participants felt that this impact could be considerably reduced through health psychology interventions. In addition to supporting their physical and mental health, the programs, led by a team of professionals (health psychologists, dieticians, and physiotherapists), provided participants with a community experience that became an experience of basic security. Furthermore, the teams’ collaboration with GPs also helped increase patient compliance (Theme 5). These experiences show that beyond the beneficial elements of the actual service, systemic aspects of integrated primary care can reach individual patients and communities and convey to them feelings of basic security.

## 5. General Discussion

In the present study, we provided quantitative and qualitative results about the monitoring of a reformulated primary care service, the Primary Health Care Development Model Program, in the most disadvantaged regions of Hungary. The four GP clusters in the Model Program provided a community and prevention-focused service that relied on various health professionals. Among them, the program employed community health psychologists committed to exposing and challenging harmful social inequalities as part of a collective action to create more health-promoting social environments for those potentially involved in social struggles [20,21,22]. Our research questions and analyses focused specifically on various aspects of the community health psychology services in the Model Program.

In line with our first research question, we tested whether the subpopulations in need had better access to health psychology services. Our findings suggest mixed results. With the targeted administration and support of the Model Program, people with disadvantaged sociodemographic and health characteristics were able to access a higher level of healthcare, including new services that were not available before, such as different forms of health psychological support. We found that these services were more likely to be accessed by those who need them the most: clients in poorer general health and those with stronger depression. However, improvements must be made to reach out to other subgroups such as those with lower education levels, male patients, and the Roma minority [48]. It is important to note here that other program elements might have reached these populations, too [28].

Our second study tested the potentially beneficial outcomes of the health psychology services for the participants. Once the patients accessed the Model Program’s individual and group health psychology services, they experienced significant mental health improvements during and after the interventions: less-severe depression and a higher level of general well-being were reported. These results support the assumption that the services provided by health psychologists benefit individual clients [49]. More specifically, individual and group interventions may be effective in poorer communities and those comprised of the Roma ethnic minority. Our results converge with previous findings, indicating that participation in individual or group health psychology interventions is potentially beneficial in primary care [33,34,35,36,37,38,39].

It is a general phenomenon worldwide that many patients with mental health issues seek help only at the stage of primary health care, and research suggests that most mental disorders are treated by general practitioners [18,50]. However, psychological interventions may reduce GPs’ burden of treating patients with psychosomatic complaints, for example, medically unexplained symptoms [51,52], which is a prevalent problem in disadvantaged populations [53]. However, the occurrence of a higher rate of depression may indicate the need for specialized psychiatric/psychotherapeutic care, which would be beyond the competence of community health psychologists working in integrated primary care.

While this research generated quantitative data about the accessibility and potential mental health benefits of health psychology services, our third research question and the subsequent qualitative investigation focused on the personal testimonies of participants. Based on the themes of the focus-group interviews, the health psychology services appear to have had fundamentally empowering effects. Respondents reported the experience of mental and behavioral resource mobilization processes in all four practice communities. There was a growing experience of perceived social support, an increase in self-confidence and social activity, and broadened horizons due to psychoeducation. These experiences can enhance physical–mental recovery, treatment adherence, and rehabilitation [54] and provide an opportunity for sharing experiences in the community [22]. Moreover, shared experiences may reduce the shame associated with illness through fostering acceptance and helping cope with loss, grief, stress, and negative feelings. The themes of the qualitative study (such as “Community instead of isolation” and “Autonomy and activity”) strengthen the claim of the need to work at individual and community levels at the same time in community health psychology [55].

### Limitations

The research presented here cannot yet answer some questions due to several limitations. First, we highlight the relatively small number of outcome-monitoring study respondents compared to the whole HSA sample, which should be considered when interpreting the results. The participant involvement process was challenging because the health psychology service-capacity development was slower than expected during the Model Program. Health psychology services are a labor-intensive form of care that reach significantly fewer participants than sizeable public health programs (e.g., information days). Consequently, the capacity of psychologists was limited, as only one psychologist per practice community was available. Additionally, because of the clinical level of mental health problems, many patients were referred to psychiatrists and could not participate in the health psychological services.

Second, in these targeted disadvantaged regions, patients’ lower level of education, socioeconomic status, lack of experience and knowledge about health psychology, and the stigma associated with seeking psychological help may have affected the willingness to engage in consultations or group sessions [25,26,27]. Finally, since the information on health psychology services was provided by GPs and the health mediators, and application to the program was voluntary, access to the service and participation in the study might have been influenced by the frequency of patients’ visits to their GPs’ surgeries.

Third, the outcome-monitoring study was only conducted in the last year of the Model Program, which means that not everyone took part in it who received health psychological services during the whole program period. Accordingly, the monitoring period was relatively short compared to the total duration of the Model Program. The results thus necessarily represent only a part of the total operating time of the Model Program, which also explains why a detailed analysis of each form of intervention was not possible. Therefore, in further developing and extending community health psychology programs, special attention must be devoted to ensuring that mental health assessments become part of services in the form of routine outcome monitoring [41,42] both during involvement in such services and during their closure and the follow-up phase.

Finally, the generalizability of the results of the outcome-monitoring study may also be limited, given that we did not involve a control group against which we could have compared the intervention group. While the results show a significant increase in participant well-being during the program, this improvement may be attributed to general methodological factors and other program elements. It is recommended that further studies use randomized controlled designs to provide evidence of the extent to which the mental health benefits can be attributed to specific health psychological interventions. In addition, participation in the intervention study was voluntary, which may have led to self-selection bias.

## 6. Conclusions

We identified from our mixed-method study that health psychology care, when durably available, was considered especially supporting and empowering in communities in disadvantaged regions of Hungary. However, participants in all communities that experienced the benefits of the health psychology services had concerns regarding the termination of the program: they desired the long-term availability of services provided by psychologists. Therefore, we consider it extremely important that community health psychologists undergo preparatory training before starting work. During training, they can formulate an idea of the unique requirements and challenges of primary care work and the specific needs of different socio-cultural groups [56,57].

The results presented here demonstrate that community health psychology services implemented at the primary health care level can contribute to improving mental health indicators and general health functioning even among patients who belong to the most disadvantaged groups. Moreover, community health psychologists can work on empowering these communities and addressing their health needs. In this way, the development of the GP clusters creates new, complex social–ecological environments. The development of these environments—also called personal niches [58]—is a systemic requirement for promoting the health of individuals and communities. More generally, the latter may develop a psychological and community relationship culture, thus reducing social inequality and increasing general well-being.

## Figures and Tables

**Table 1 ijerph-20-03900-t001:** Predictors of the receipt of health psychology services through the model program.

				95% C.I. to the OR	
	B	*p*	OR	Lower	Upper
Gender: female (reference: male)	1.104	<0.001	3.015 ***	1.823	4.987
Age (years; reference: 18–24)		0.182			
25–44	0.337	0.403	1.401	0.636	3.086
45–64	0.008	0.985	1.008	0.440	2.310
65+	−0.388	0.425	0.679	0.262	1.759
Education (reference: maximum 8 years of primary school)		<0.001			
Secondary without graduation	1.467	<0.001	4.337 ***	2.023	9.299
Secondary with graduation	1.892	<0.001	6.632 ***	3.252	13.523
Higher education	2.101	<0.001	8.172 ***	3.655	18.272
Perceived/subjective financial situation (reference: bad/very bad)		0.450			
Optimal	0.339	0.207	1.403	0.829	2.375
Good/excellent	0.306	0.413	1.358	0.652	2.829
Roma identity: yes (reference: no)	−1.223	0.231	0.294	0.040	2.177
BMI (reference: no overweight)		0.701			
Overweight	−0.200	0.433	0.819	0.497	1.349
Obese	−0.033	0.894	0.967	0.594	1.574
SRH (reference: bad/very bad)		0.014			
Optimal	0.264	0.380	1.301	0.723	2.343
Good/excellent	−0.493	0.192	0.611	0.292	1.280
How much can you do for your health? (reference: I can do very much)		0.146			
I can do much	−0.566	0.049	0.568	0.323	0.998
I can do little	−0.658	0.083	0.518	0.246	1.089
There’s nothing I can do	−1.686	0.112	0.185	0.023	1.482
Systolic blood pressure: high (reference: normal)	0.429	0.085	1.536	0.943	2.502
Diastolic blood pressure: high (reference: normal)	−0.017	0.950	0.983	0.581	1.663
Smoking (reference: not smoking)	−0.298	0.208	0.743	0.467	1.180
BDI-S (reference: 0–9: normal)		0.002			
Mild depression (10–18)	0.257	0.384	1.293	0.725	2.305
Moderate depression (19–24)	1.203	0.001	3.329 **	1.660	6.678
Severe depression (25+)	1.307	0.002	3.694 **	1.618	8.430
GHQ12: high (5+) (reference: 0–4: normal)	0.773	0.015	2.167 *	1.165	4.029
Constant	−7.321	<0.001	0.001		

Notes: SRH, self-rated health; BDI-S, Beck Depression Inventory—Short Version; GHQ12, General Health Questionnaire 12. *: *p* < 0.05; **: *p* < 0.01; ***: *p* < 0.001.

**Table 2 ijerph-20-03900-t002:** Descriptive characteristics of the outcome-monitoring study’s sample.

	Measurement Point	
	T1	T2
N	156	137
Male	23	22
Female	133	115
Age	47.5 (15.9)	47.8 (16.0)
Individual	81	67
Group	75	70
T2–T1 (day)		119.4 (79.8)

Notes: missing values are not shown.

**Table 3 ijerph-20-03900-t003:** Comparison of variables included in the intervention outcome-monitoring study.

	T1		T2		t-Rest	*p*	F-Test^+^	*p*	eta^2^
Variable	M	SD	m	SD					
BDI-S	17.90	5.85	13.69	3.84	9.27	<0.001	4.52(131)	0.035	0.033
WHO-WBI	6.63	3.22	8.96	2.63	−7.51	<0.001	2.32(131)	0.130	0.017

Notes: one-way repeated measure ANCOVA controlled for gender, age, education level, and BMI; BDI-S, Short Beck Depression Inventory; WHO-WBI, WHO Well-Being Index.

## Data Availability

The data presented in this study are available on request from the corresponding author. The data are not publicly available due to privacy restrictions.

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
