# Peer review of "Health Psychology Services for People in Disadvantaged Regions of Hungary: Experiences from the Primary Health Care Development Model Program"

_ijerph, 2023, doi:10.3390/ijerph20053900_

Round 1
Reviewer 1 Report
This manuscript presented a mixed-method study monitoring the efficacy of health psychology services. They conducted three studies, and some analysis methods, such as multivariate binary logistic regression analysis, etc., have been applied to the health psychology data referring to questionnaire, focus group interviews, etc. However, this manuscript cannot give some new perspective, the analysis methods are too general and simple without any theoretical presentation. The experiments just show the objective results without measurable and comparative analysis.
Reviewer 2 Report
The article of "Health psychology services for People in Disadvantaged Regions of Hungary: Experiences from the Primary Care Development Model Program" is showing a fundamental study about a pilot project placed in Hungary. The importance of the study is inevitable. The topic of the study is timely, relevant, significant and interesting.
The Introduction includes the most important references and showing the Hungarian context and the background literature properly. The study uses sufficient theoretical constructs. The main aim of the study is to examine the characteristics and effectiveness of the health psychology services of the Primary Care Development Model Program in Hungary. The aims of the study are clear and significant, however, it is complex and challenging to study those aims. To study the efficacy of an intervention is multifaceted and many risk lies in the natural setting. The researcher must be very cautious to draw conclusions and show real correlations. The authors are self-reflective about this matter.
The researchers chose the mixed-methods study which is effectively combining both qualitative and quantitative research methods to provide a comprehensive understanding of the topic at hand. The study begins with a quantitative survey (Study 1, Study 2) to gather numerical data on the subject, followed by focus group interviews (Study 3) to provide qualitative insights and personal perspectives.
The strength of the study are that the sample size of the survey, and the consciously chosen sample of the focus group interviews, which can affect the results. They used rigorous sampling procedures. The ’health mediators’ seems to play an important role in the recruitment, however, their role and tasks are not clearly defined. This could be clarified.
The survey data provides a broad overview of the topic, while the focus groups provide detailed and nuanced information. This allows for a well-rounded and holistic understanding of the subject, this could be elaborated more. The study is well-designed with clear objectives and research questions, which are answered mainly separated in the studies. The studies have a clear and concise reporting of results, making it easy for readers to follow and understand the key findings. However, a little more understanding of how these results relating to each other would help to reach an excellent study altogether. One weakness of the study is the way in which the quantitative and qualitative data are integrated and used to complement each other.
Rethinking the usage of the term „efficacy” is also recommened. It stays a huge question in such studies whether it is the efficacy of the intervention or what else could be intermediating.
Overall, this mixed methods study provides a valuable contribution to the field by effectively using both quantitative and qualitative research methods to gain a comprehensive understanding of the topic. The study is well-designed and executed, and the results are clearly reported.
Round 2
Reviewer 1 Report
This manuscript has some improvements.